TOPICAL REVIEW

# Motor unit adaptation to disuse: crossing the threshold from firing rate suppression to neuromuscular junction transmission

Mathew Piasecki 

*Centre of Metabolism, Ageing & Physiology (CoMAP), Medical Research Council/Versus Arthritis UK Centre of Excellence for Musculoskeletal Ageing Research (CMAR), NIHR Nottingham Biomedical Research Centre, University of Nottingham, Derby, UK*

Handling Editors: Laura Bennet & Martino Franchi

The peer review history is available in the Supporting Information section of this article (https://doi.org/10.1113/JP284159#support-information-section).

**Abstract figure legend** Neuromuscular disuse scenarios of limb immobilization, reduced activity and bed rest result in impairments of muscle strength that exceed losses of muscle size. Neural adaptations are an assured consequence of disuse. The available evidence garnered from human and animal models highlights suppressed motor unit firing rate (MUFR) that is more evident in lower-threshold motor units, in which electrophysiological methods indicate no discernible impairment of the neuromuscular junction (NMJ). Numerous other methods corroborate NMJ impairment following disuse, and this may be applicable only to higher-threshold motor units. The dashed arrow indicates the theoretical and simplified relationship between motor unit recruitment and muscle force generation.

**Mathew Piasecki** is an Associate Professor within the Centre of Metabolism, Ageing and Physiology (COMAP) at the University of Nottingham, UK. His research interests are focused on the neural input to muscle in ageing and disease and how decrements of this might be alleviated with intervention.

**Abstract**  Neural conditioning to scenarios of muscle disuse is undoubtedly a cause of functional decrements that typically exceed losses of muscle size. Yet establishing the relative contribution of neural adaptation and the specific location in the motor pathway remains technically challenging. Several studies of healthy humans have targeted this system and have established that motor unit firing rate is suppressed following disuse, with a number of critical caveats. It is suppressed in the immobilized limb only, at relative and absolute force levels, and preferentially targets lower-threshold motor units. Concomitantly, electrophysiological investigation of neuromuscular junction transmission (NMJ) stability of lower-threshold motor units reveals minimal change following disuse. These findings contrast with numerous other methods, which show clear involvement of the NMJ but are unable to characterize the motor unit to which they belong. It is physiologically plausible that decrements observed following disuse are a result of suppressed firing rate of lower-threshold motor units and impairment of transmission of the NMJ of higher-threshold motor units. As such, motor units within the pool should be viewed in light of their varying susceptibility to disuse.

(Received 7 February 2024; accepted after revision 21 October 2024; first published online 4 November 2024)

**Corresponding author** Mathew Piasecki: Centre of Metabolism, Ageing & Physiology (CoMAP), Medical Research Council/Versus Arthritis UK Centre of Excellence for Musculoskeletal Ageing Research (CMAR), NIHR Nottingham Biomedical Research Centre, University of Nottingham, Derby DE223DT, UK.   Email: mathew.piasecki@nottingham.ac.uk

## Introduction

Experimental approaches to physiological disuse have primarily explored underlying mechanisms of reduced neuromuscular function and muscle mass. These studies have translational implications for clinical settings, spaceflight and, in some instances, are used as a model of accelerated ageing (Deane et al., 2024). Methods of intervention differ in their approach and range from full bed rest to explore effects of whole-body disuse to individual limb immobilization or suspension (Atherton et al., 2016). Studies of individual limb disuse are arguably more tolerable for participants and ethically agreeable, and although this limits the effects of more systemic disuse, it does enable the exploration of muscle- and limb-specific adaptation, with the benefit of using non-immobilized opposing limbs as a comparison (Preobrazenski, Janssen et al., 2023). Crucially, immobilization studies also differ in duration, and collective evidence suggests a rapid decline in function, with a plateau during prolonged disuse (Preobrazenski, Seigel et al., 2023). Several outcomes are unquestionable; disuse results in a marked reduction in muscle size, which is exceeded by a reduction in muscle strength and power (Campbell et al., 2019; Hardy et al., 2022; Marusic et al., 2021). This latter factor highlights a potent influence on the neural input to muscles, an area that has garnered comparatively less research attention than the molecular mechanisms of muscle contraction.

Of the data available in humans and animals, a reduction of motor unit firing rate (MUFR) and disruption at the neuromuscular junction (NMJ) are clear consequences of disuse, but the underpinning mechanisms are less clear. To identify potential impairments in the process, it is essential to understand fully the complex factors governing motor unit (MU) activation and firing, in addition to effective communication from nerve to muscle.

## Motor pathway

The MU is the final element of the motor pathway, consisting of an $\alpha$-motoneuron and its axon, numerous NMJs and all the individual muscle fibres they innervate. The amount of force generated with a voluntary contraction is contingent upon the recruitment of MUs and the rate at which they discharge action potentials (i.e. the MUFR). Recruitment of an MU is dependent on its activation threshold, which is typically proportional to its size. Smaller motoneurons with higher input resistance reach activation threshold with lower synaptic input compared with larger motoneurons. Smaller MUs are typically composed of fewer and smaller muscle fibres (type I) and generate lower levels of force than larger fibres (type II) (Enoka & Duchateau, 2017). Both recruitment and MUFR are altered to suit the desired level of force for all movements and interactions with the environment, and this can be described with a series of steps, each of which can be viewed as an independent research field and is described briefly herein.

In response to numerous stimuli, such as sight, sound and afferent feedback from proprioceptive impulses, signals from the primary motor cortex (M1) and reticular

formation of the brainstem propagate down descending corticospinal and reticulospinal tracts, where they are, in turn, transmitted to interneurons and motoneurons (Glover & Baker, 2022; Kiehn, 2016). The M1 region in humans and higher primates differs from other animal models and has distinct evolutionary origins. Subdivided into a rostral region (old M1) and a caudal region (new M1), the old M1 might have fewer or more slowly conducting cortico-motoneuronal cells and communicates with lower motoneurons via spinal interneurons (Witham et al., 2016).

Of those descending cells that do not synapse directly onto motoneurons, groups of excitatory and inhibitory interneurons regulate the net excitability of motoneurons (Zholudeva et al., 2021). In bilateral movements, the regulation of each limb is partly reliant on the prevention of contralateral motoneuron excitation, referred to as crossed inhibition. This can occur directly via inhibitory commissural neurons (CNs) or indirectly via excitatory CNs acting on premotor inhibitory neurons. Cross-excitation can also occur via direct CN excitation of interneurons/motoneurons (Maxwell & Soteropoulos, 2020) and is a prominent consideration for studies of unilateral disuse, where detrimental effects can be expected in the non-immobilized limb.

The net excitation of the motoneuron can occur from thousands of synaptic inputs along its dendrites, ranging from descending drive to sensory afferents, all of which must be integrated and regulated. This is achieved via two fundamental mechanisms: ionotropic input and neuromodulation. Postsynaptic ionotropic receptors depolarize or hyperpolarize the cell via regulation of ion exchange across the membrane to generate an excitatory or inhibitory response (Heckman et al., 2009). The monoamines serotonin (5-HT) and noradrenaline (NA) are potent neuromodulators released from caudal raphe neurons and the locus coeruleus, respectively. These neurotransmitters stimulate persistent inward currents (PICs) via G-protein-coupled second messengers, which act to amplify and prolong synaptic input to the motoneuron and enable an input–output gain control of motoneurons that facilitates motor output in response to demand (Johnson & Heckman, 2014). Direct estimation of PIC amplitude in human spinal motoneurons is not possible, but the well-established $\Delta F$ technique, in which MUFR onset–offset hysteresis is calculated during ramped voluntary contractions, is able to estimate the influence of neuromodulatory inputs on the amplification of synaptic input (Gorassini et al., 1998; Mesquita et al., 2024).

Much of the net excitation is common across multiple motoneurons (Deluca & Erim, 1994), and monoaminergic drive is highly diffuse (Johnson & Heckman, 2014), meaning that the selective activation of individual motoneurons is improbable (Rossato et al., 2024). Once depolarized, successive action potentials propagate along the motoneuron axon and axonal branches to terminate at NMJs. The NMJ has a highly specific role in the transmission of ACh to the postsynaptic motor endplate and initiates a muscle fibre action potential. Indeed, this 1:1 ratio of a motoneuron and muscle fibre action potential has enabled an active field of research; with appropriately placed recording electrodes, action potentials are readily detectable from the muscle and reveal structural and functional characteristics of the MU (Del Vecchio et al., 2020; Jones et al., 2021). There are scant combined histological and electrophysiological data on the NMJ from humans, and the relationship between the structural appearance of the NMJ, often termed stability, and the function of the NMJ, its transmission (in)stability, has not been associated reliably. Put simply, an NMJ of non-conventional appearance might well be able to continue to regulate ACh release effectively.

## Suppression of motor unit firing rate

Following 15 days of leg immobilization in healthy young humans, vastus lateralis muscle size reduced by ~15%, which was far exceed by a reduction in strength of the knee extensors of ~31% (Inns et al., 2022). The MUFR recorded at submaximal contractions (10% and 25% maximum) was also reduced after disuse by ~10%. Given the pivotal role of MUFR in force generation, this suppression of MUFR after disuse might initially be viewed as a consequence of sampling MUs at reduced force levels rather than a cause of lower force generation. Yet the same pattern of suppressed MUFR was apparent when contraction levels were normalized to baseline maximum strength (prior to loss of force), favouring suppressed MUFR as a direct consequence of disuse (Inns et al., 2022).

In a similar yet independent study, following 10 days of lower limb suspension in healthy young males the vastus lateralis size was reduced by ~4.5% and, again, was far exceed by a ~30% reduction in strength (Sarto et al., 2022; Valli et al., 2023). The MUFR during sustained contractions also decreased at 10% and 25% maximum voluntary contraction (MVC) by 12% and 10%, respectively. Importantly, this was threshold level specific, with MUFR at 50% MVC showing a ~6% increase (Valli et al., 2023). Preceding these studies by decades and underpinning their findings, Duchateau and Hainaut (1990) immobilized the hand for 6–8 weeks and reported a reduced maximal MUFR in the adductor pollicis and first dorsal interosseous muscles of ~41%. This suppression also disproportionately affected low-threshold MUs. Likewise, a decrease of ~31% and 37% of MUFR was reported following 3 and 6 weeks, respectively, of immobilization of the first dorsal interosseous (Seki et al., 2001). Similar to the unquestionable decline in strength, the accumulated evidence in humans

 M. Piasecki 

strongly suggests that suppressed MUFR plays a causal role in the loss of strength induced by disuse, and this appears to be targeted preferentially towards earlier recruited lower-threshold MUs. However, the specific factor causing the suppression of MUFR remains less evident. At the simplest level, suppression of MUFR in these studies can be viewed as an alteration in the gain control of the motoneuron and/or an altered balance of excitation and inhibition of motoneurons, inclusive of the intrinsic properties of motoneurons; either ionotropic, neuromodulatory, or both (Johnson & Heckman, 2014).

Consider an opposing stimulus to disuse, that of resistance and/or endurance exercise training. Here, convincing evidence highlights central neural adaptation as a cause of improved performance (Pearcey et al., 2021), including within intracortical circuits (Glover & Baker, 2020) and descending drive (Glover & Baker, 2022). However, the temptation to view these opposing interventions as having directly opposing effects should be resisted; the commonly reported cross-education effect noted with unilateral exercise (Altheyab et al., 2025; Calvert & Carson, 2022), whereby an opposing non-exercised limb also improves, has no opposing equivalent with disuse, because no decrements of strength or MUFR were noted in control limbs (Duchateau & Hainaut, 1990; Inns et al., 2022; Preobrazenski, Janssen et al., 2023). This was apparent in control muscles of the lower limb and hand, indicating no bilateral effect (or 'cross-uneducation') to muscles served predominantly via interneurons or directly via cortico-motoneuronal connections. This is further supported by functional MRI data showing decreased activity in regions contralateral to the immobilized limb only, alongside reduced corticospinal excitability, with no ipsilateral adaptation following unilateral immobilization (Avanzino et al., 2011; Garbarini et al., 2019). Collectively, it seems that impaired cortical activity and descending drive are probable contributors to disuse-induced neuromuscular impairment, and this is specific to the immobilized limb.

From a positive perspective, vastus lateralis MUFR at normalized contraction levels returned to baseline following a short resistance exercise training rehabilitation programme (Sarto et al., 2022; Valli et al., 2023), and in first dorsal interosseous, deficits observed after 6 weeks of recovery were completely restored with an additional 6 weeks of non-interventional recovery (Seki et al., 2001). These combined findings strongly indicate that conventional intervention methods, such as resistance exercise training or even normal daily activity, effectively restore MUFR after a period of disuse, albeit with current evidence restricted to healthy young people.

Recall that monoaminergic (e.g. 5-HT and NA) drive to spinal motoneurons has potent effects on the intrinsic excitability of the motoneuron via PIC activation, which amplifies synaptic input (Heckman et al., 2009). The onset–offset hysteresis of motoneuron firing as assessed by $\Delta F$ is highly adaptable, with increases in response to increased activity (Orssatto, Blazevich et al., 2023; Orssatto, Rodrigues et al., 2023), decreases in response to induced acute inhibition (Mesquita et al., 2022; Orssatto et al., 2022), and is markedly lower in young males compared with females (Jenz et al., 2023) and in older compared with young subjects (Guo, Jones, Škarabot et al., 2024; Hassan et al., 2021; Orssatto et al., 2021). In extended analysis of data from the aforementioned 10 day lower limb suspension study (Sarto et al., 2022; Valli et al., 2023), $\Delta F$ from trapezoidal ramped contractions reduced immediately following immobilization and recovered following resistance exercise training (Martino et al., 2024). The degree of adaptation of $\Delta F$ was correlated with MVC fluctuations over this period, suggesting an influence of disuse on PIC-mediated sustained MU firing (Martino et al., 2024). However, $\Delta F$ scales non-linearly with contraction intensity in the vastus lateralis (Škarabot et al., 2023), and lower normalized values following strength loss might be a consequence of the reduced force level at which it was calculated.

If the PIC contributions to motoneuron firing are altered following disuse, further questions remain regarding the causative mechanism; is it reduced availability of 5-HT and NA, and/or a reduction in mono-amine receptors or their impairment, or an altered balance of excitatory and inhibitory inputs? All have physiological plausibility and, to some extent, supporting evidence from animal models. In rats, results of 5-HT activity following short periods of immobilization are mixed, showing increases (Takahashi et al., 1998) and decreases (Clement et al., 1998). But, used as a model of stress inducement, these brief periods of inactivity are unlikely to translate directly to humans and might not reflect true effects of immobilization. Furthermore, there is a strong likelihood that 5-HT neuronal activity is positively correlated with motor output, as shown in cats (Jacobs et al., 2002). Considering that MUFR in humans were suppressed at multiple force levels (normalized to relative and absolute maximal) after disuse (Inns et al., 2022), it is possible that descending drive did not increase when greater relative effort was required or that motoneurons were less responsive to it. Notwithstanding this reduced MUFR, these forces after disuse were achievable and, presumably, were facilitated via greater MU recruitment; however, current methods limit the reporting of MU behaviour to only those which can be sampled.

There is also limited evidence to directly indicate reduced or impaired receptors on motoneuron dendrites following disuse. Indeed, the opposite might be more likely, because synaptic scaling (the upscaling of post-synaptic strength in response to reduced synaptic input) is applicable to motoneurons (Santin et al., 2017). This might partly explain the rapid recovery of the vastus

lateralis (Martino et al., 2024; Sarto et al., 2022; Valli et al., 2023) and first dorsal interosseous (Seki et al., 2001) when activity resumed after disuse.

A critical point of the available human data is that the reduction in MUFR is threshold specific and preferentially affects lower-threshold MUs (Duchateau & Hainaut, 1990; Valli et al., 2023). In response to pain, the firing rates of lower-threshold (20% MVC) tibialis anterior MUs were suppressed, and firing rates of higher-threshold (70% MVC) MUs were increased (Martinez-Valdes et al., 2020), which suggests a stronger inhibitory influence on the lower-threshold MUs. Supporting this increased inhibition hypothesis from a mechanistic perspective, 2 weeks of cast immobilization in rats induced hyperalgesia (Ohmichi et al., 2012); prolonged fixed joints, such as adopted with disuse, could reasonably alter muscle spindle activity (Lan & He, 2012), and the lack of contraction-induced deformation of afferents might render them hypersensitive with reuse. Furthermore, disuse-associated inflammation might also have inhibitory effects via group III/IV inhibitory afferents (Amann, 2012; Jones et al., 2023).

The collective evidence points to suppression of MUFR with disuse in humans, with several caveats: (i) it is specific to the limb immobilized and has no negative cross-over effects ('cross-uneducation'); (ii) it remains suppressed in comparison to baseline when assessed at relative and absolute contraction forces; and (iii) it is more apparent in lower-threshold MUs.

## Transmission at the NMJ

The NMJ is the final point of the motoneuron, bridging nerve–muscle synaptic communication to initiate muscle fibre action potentials and contraction. Much of what is known about the NMJ stems from animal models, but exploration in humans is possible with electrophysiology (Piasecki et al., 2021) and immunohistochemistry (Boehm et al., 2020) techniques. Intramuscular electromyography during voluntary contractions in humans enables the quantification of consecutive motor unit potentials (MUPs) and their near-fibre potentials across multiple depths of muscle (Jones et al., 2021; Piasecki & Stashuk, 2023). Trains of MUPs are used to generate a MUP template, features of which can be used to infer anatomical features of the MU (Jones et al., 2022). High-pass filtering of the MUP and corresponding train generates near-fibre MUPs (Stashuk, 1999), and the variability across consecutive observations can be used to infer the variability of NMJ transmission times (Piasecki et al., 2021). Termed NMJ transmission instability, this assessment relates to the dynamics of ACh release and postsynaptic binding, which is greater than the variability observed in action potential conduction velocity of

axonal branches and muscle fibres, hence it has a greater influence on near-fibre MUP shape (Katz & Miledi, 1965).

This method was used in two independent studies of leg immobilization in healthy young males, and both reported no change in NMJ transmission instability after 10 (Sarto et al., 2022) and 15 days of disuse (Inns et al., 2022). It is important to note limitations here. These NMJs were assessed at fairly low contraction levels (∼25% of maximum) and reveal little of the NMJs of higher-threshold MUs, where alternative imaging methods might be more advantageous.

Histological imaging of the human NMJ, particularly the presynaptic region, poses significant technical challenges. The most detailed data often come from amputated limbs, where access to full longitudinal sections of muscle is possible (Boehm et al., 2019; Jones et al., 2017). However, targeted human biopsy techniques do improve NMJ yield beyond standard techniques (Aubertin-Leheudre et al., 2020). Direct histochemical imaging of the NMJ structure in young maturing rodents following 10 weeks of limb suspension showed a marked decrease in the size of the postsynaptic motor endplate, with no change to presynaptic regions (Deschenes et al., 2006). Similar effects on the postsynaptic endplate were observed after only 10 days of unloading (Deschenes et al., 2005), but no effect was observed in fully mature rodents (Deschenes & Wilson, 2003). This form of structural NMJ adaptation with disuse is age specific and more evident in those still undergoing development (Deschenes et al., 2021). Nevertheless, the importance of the methods of immobilization/disuse are underscored using the rodent model, because pharmacological inhibition of the NMJ had opposing effects to the prevention of motoneuron activation via spinal hemi-section (Mantilla et al., 2007); blocking of the NMJ resulted in a reduced size of the synaptic vesicle pool, which was increased when the motoneuron soma was blocked (Mantilla et al., 2007). This work shows that motoneuron activation, which is difficult to quantify in human disuse studies, clearly influences peripheral NMJ adaptation.

More indirect imaging methods point to disuse-induced NMJ disruption in humans. Neural cell adhesion molecule (NCAM) is a muscle fibre cell surface protein used as one of several molecular markers to highlight innervation status, with NCAM$^+$ fibres assumed to be denervated (Soendenbroe et al., 2021). The proportion of NCAM$^+$ fibres in human vastus lateralis increased after 3 (Demangel et al., 2017), 10 (Monti et al., 2021) and 14 days of bed rest (Arentson-Lantz et al., 2016). RNA-sequencing analysis also highlighted NMJ-specific alterations with 10 days of limb suspension (Sarto et al., 2022); single-fibre NCAM expression was increased following spaceflight (Murgia et al., 2022), and a number of potential circulating biomarkers specific to the NMJ have been proposed (Sirago et al., 2023).

Several lines of human evidence call attention to NMJ disruption with neuromuscular disuse, but a functional impairment of the NMJ was not detectable with electrophysiological methods at low- to mid-level contractions. A possible notable caveat of this process is the susceptibility of some NMJs of higher-threshold MUs to undergo disruption while those of lower-threshold remain unaltered. Although neuromuscular effects of ageing are not directly comparable to short-term disuse, some similarities exist. In older rats, faster muscle fibres (presumably belonging to higher-threshold MUs) are more susceptible to NMJ disruption and denervation than slow muscle fibres (Kadhiresan et al., 1996). The reasons for this susceptibility might be related to differences in NMJ structure; when compared with type II fibres, NMJs innervating type I fibres are smaller, have greater overlap between pre- and postsynaptic regions, and are less fragmented (Sieck & Prakash, 1997). Type II (IIx and IIb in rodents) NMJs are also more susceptible to transmission failure (Deschenes et al., 1994; Sieck & Prakash, 1997).

## Conclusion and future directions

It is probable that the effects of disuse on MUs are not equally applied to all MUs, with a suppression of MUFR that affects lower-threshold MUs disproportionally, and with NMJ disruption affecting higher-threshold MUs disproportionately. The two processes are not at odds, and it is feasible that they are mutually explanatory; the increase in MUFR of higher-threshold MUs used during higher contractions (50% MVC; Valli et al., 2023) might act to overcome impaired muscle contraction caused by NMJ disruption, which does not occur in lower-threshold MUs. This is somewhat of a simplification, but the forces at which MUs are sampled should be a key consideration for future mechanistic studies in this field.

Molecular aspects of spinal motoneurons in humans will probably continue to elude us owing the huge methodological constraints that prohibit their investigation *in vivo*. Nonetheless, recent advances in both EMG hardware and decomposition techniques are enabling a greater number of MU spike trains and, more importantly, from a greater range of contraction intensities, to be sampled simultaneously across the volume of muscle (Avrillon et al., 2024; Chung et al., 2023; Škarabot et al., 2023).

Of the four human cohorts covered herein in which individual MUs were sampled, 32 of the 33 combined participants were male. This limitation might be problematic given the documented differences in male and female MU function (Guo et al., 2022; Guo, Jones, Smart et al., 2025; Jenz & Pearcey, 2022; Lulic-Kuryllo & Inglis, 2022), and any further physiological constraints of including females appear minimal, given that similar methods of interrupted disuse have been used in female-only cohorts (MacLennan et al., 2021).

Finally, although useful from a mechanistic standpoint, the range of laboratory-based assessments performed to date is somewhat limited. Unilateral MU behaviour during submaximal isometric contractions might reveal little of motor pathway commands during more dynamic functions or those applicable to activities of daily living, such as bilateral MU function during normal gait. This dynamic assessment is methodologically challenging but may generate a research capability in which the MU pool is viewed not as single entity, but as a range with differing susceptibility to intervention.

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

## Additional information

### Competing interests

The author has no competing interests to declare.

### Author contributions

Sole author.

### Funding

None.

### Acknowledgements

I am grateful to several colleagues for the many discussions around neuromuscular function in health, disease and disuse.

### Keywords

disuse, firing rate, motoneuron, neuromuscular junction

### Supporting information

Additional supporting information can be found online in the Supporting Information section at the end of the HTML view of the article. Supporting information files available:

**Peer Review History**

