## [Peer Review History · The Journal of Physiology]

Motor unit adaptation to disuse: crossing the threshold from firing rate suppression to neuromuscular junction transmission

Mathew Piasecki

DOI: 10.1113/JP284159

Corresponding author(s): Mathew Piasecki (mathew.piasecki@nottingham.ac.uk)

The following individual(s) involved in review of this submission have agreed to reveal their identity: C. J. Heckman (Referee #2)

Review Timeline:

Submission Date:	07-Feb-2024
Editorial Decision:	15-Mar-2024
Revision Received:	02-Aug-2024
Editorial Decision:	27-Aug-2024
Revision Received:	09-Oct-2024
Accepted:	21-Oct-2024

Senior Editor: Laura Bennet

Reviewing Editor: Martino Franchi

Transaction Report:

Dear Mathew,

Re: JP-TR-2024-284159 "Neural conditioning to muscle disuse: crossing the threshold from firing rate suppression to neuromuscular junction transmission" by Mathew Piasecki

Thank you for submitting your manuscript to The Journal of Physiology. It has been assessed by a Reviewing Editor and by 2 expert referees and we are pleased to tell you that it is potentially acceptable for publication following satisfactory major revision.

Please address all the points raised and incorporate all requested revisions or explain in your Response to Referees why a change has not been made. We hope you will find the comments helpful and that you will be able to return your revised manuscript within 2 months. If you require longer than this, please contact journal staff: jp@physoc.org. Please note that this letter does not constitute a guarantee for acceptance of your revised manuscript.

REVISION CHECKLIST:

We look forward to receiving your revised submission.

Best wishes,

Laura Bennet
Senior Editor
The Journal of Physiology

REQUIRED ITEMS

- Please upload separate high quality figure files via the submission form.

- Author profile(s) must be uploaded via the submission form. Authors should submit a short biography (no more than 100 words for one author or 150 words in total for two authors) and a portrait photograph of the two leading authors on the paper. These should be uploaded and clearly labelled together in a Word document with the revised version of the manuscript. Any standard image format for the photograph is acceptable, but the resolution should be at least 300 DPI and preferably more. A group photograph of all authors is also acceptable, providing the biography for the whole group does not exceed 150 words.

EDITOR COMMENTS

Reviewing Editor:

The commissioned review has now been revised by two reviewers who are very expert in the field of neuromuscular control of movement. Both reviewers highlighted pros and cons of the present article. Specifically, Reviewer 1 has provided an interesting and on-point (IMHO) comments related to the conclusions made by the author on how disuse depresses firing rate in low-threshold motor units, whereas it impairs NMJ transmission in high-threshold motor units: "There does not appear to be enough data to conclude that the impact of disuse on NMJs is greater for high-threshold motor units." Further, Rev1 raised some concerns on the quality and clarity of the writing and on the confidence in the conclusions.

The author is invited to carefully address these comments in the next draft of the manuscript.

REFEREE COMMENTS

Referee #1:

The purpose of the topical review is to discuss the role of adaptations in the firing rate of motor units and neuromuscular propagation in response to exposure to reduced levels of physical activity. The review focuses on the differences in the adaptations experienced by low- and high-threshold motor units. The conclusion is that disuse depresses firing rate in low-threshold motor units, whereas it impairs NMJ transmission in high-threshold motor units.

General Comments

1. The impact of disuse, such as that associated with leg unloading, varies across muscle and muscle fiber types (see Figure 1 in Fitts et al. JAP 89: 823, 2000).

2. The brief section on NMJs seems to focus mainly on the anatomical adaptations with little information on NMJ, such as that

indicated by the stability (e.g., jitter) of neuromuscular propagation. There does not appear to be enough data to conclude that the impact of disuse on NMJs is greater for high-threshold motor units.

Specific Comments

35 Why is it not reasonable to consider the motoneuron pool as a single physiological entity with varying susceptibility to plasticity among its members? For example, low- vs high-threshold motor units (e.g., Figure 5 in Duchateau & Hainaut, 1990; Figure 5 in Comery et al. *J Physiol* 568: 841, 2005).

39 What are the black arrows intended to indicate?

58 Contralateral connections within the nervous system suggest that the non-involved limb should not be used as a control (Manca et al. *Sports Med* 51: 11, 2021).

80 What makes the steps intricate?

82 Don't ignore the influence of brainstem pathways on the excitability of motoneurons (Glover & Baker, *J Neurosci* 42: 3150, 2022).

92 Explain the function served by the two sets of projections.

95 What aspects of coordination do they control?

96 This paragraph is difficult to read. Perhaps begin the paragraph with a sentence that states the topic to be discussed?

104 But some neurotransmitters (e.g., serotonin and noradrenaline) attach to metabotropic receptors.

107 Neuromodulatory inputs can influence most electrophysiological properties of neurons; that is, more than just recruitment threshold.

110 Discuss the influence of these neurotransmitters in terms of gain control (Johnson & Heckman *Front Neural Circuits* 8: 81, 2014; Johnson et al. *J Neurophysiol* 118: 520, 2017).

113 The ΔF technique provides a measure of the influence of neuromodulatory inputs on sustained firing and not PIC amplitude (Afsharipour et al. *J Neurophysiol* 124: 63, 2020).

116 Define cluster.

123 Del Vecchio

131 Cite the work of Vila-Chã et al. (*J Appl Physiol* 109: 1455, 2010).

133 Replace "suppression" with "depression" throughout the manuscript.

136 ...cause of the intervention?

150 Alternatively, the depression in MUFR was probably related to the extent the habitual level of activity was reduced by the intervention. For analogous interpretation, see the Figure 6A in Farina et al. (*J Neurophysiol* 101: 350, 2009) on the adjustments in discharge rate during fatiguing contractions.

168 But Pearce et al. (*Scand J Med Sci Sports* 23: 740, 2013) found a decrease in the strength of arm muscles in the contralateral limb after 3 weeks of unilateral immobilization.

180 But all these studies incorrectly used the ΔF technique to estimate PIC amplitude. To be more precise, these studies show the influence of an intervention on sustained firing.

185 Humans only?

186 Perhaps phrase this sentence on terms of the control strategy proposed by Johnson et al. (2017)?

196 Does this mean that monoaminergic drive is distributed uniformly to each motoneuron pool?

202 Couch this in terms of the decline in habitual levels of activity.

205 These parallels are questionable.

- 207 Not if the issue is presented as a decline in habitual levels of activity.
- 209 This comparison seems inappropriate.
- 218 Alternatively, the greater impact on low-threshold motor units may reflect a greater relative loss of excitatory synaptic inputs.
- 241 Define nerve:muscle size ratio.
- 243 Does disruption refer to anatomical properties?
- 244 How is the potency of a disruption quantified?
- 245 What is the translational relevance?
- 247 Explain inhibition of MN activation.
- 249 ...was increased size...?
- 250 How does this work highlight the influence of the level of synaptic input?
- 255 Which muscle?
- 269 But the described adaptation in NMJ were not evaluated relative to the recruitment thresholds of motor units.
- 271 Is there any evidence of such a feedback pathway?
- 272 susceptibility?
- 273 How is the force generated by a single motor unit influenced by that contributed by other motor units?
- 275 What is meant by the term "overlap of thresholds of MUs"?
- 281 These studies do not examine molecular aspects of spinal MNs.
- 291 Limited rather than rigid.
- 292 Dynamic instead of complex.
- 293 Inhibition and excitation of what?
- 295 Define computational outputs.

Referee #2:

This review addresses a timely and much needed overview of an important topic, changes in neural drive to motoneurons in disuse. It effectively presents a wealth of important results in a thoughtful and thought-provoking manner. I expect it will have a major influence on this field, especially as recent results are really transforming our understanding.

The initial introductory material, however, up to line 127, really needs to be reworked in terms of being more concise and worded more carefully. Once the focus shifts to reviewing recent studies, the wording and phrasing are well done. For the first part, I have noted a few instances of lack of clarity below, but there are certainly others.

Much of the initial overview of how motoneurons and MU units function is close to textbook information. This section is crucial, but perhaps review articles would generally be preferable citations instead of single examples of recent studies.

Line 50: "have occurred" is not needed in this sentence, how about "..disuse have primarily explored.."

Line 51: "This" refers to what exactly in the previous sentence? Perhaps "This focus..."

Line 65: MUFR is not defined until line 75

Line 76: " its...." should be "their", or MU singular

Line 83: Implying M1 mainly projects to the spinal CPG is odd at best. M1 projects to many types of interneurons and, in primates, directly to motoneurons. The author is clearly aware of this from comments farther down.

Line 84: "This.." seems to refer to locomotor centers

Line 94: "..those interrupted..." is not an appropriate way to describe corticospinal inputs to multiple types of interneurons

Line 109: " in this context" is unnecessary wordiness

Line 117: while Hug et al study is beautiful work establishing "clusters", the study of common drive has a long history and at least a few more references are needed here.

Line 131: why not stick with the MUFR abbreviation?

Lines 133-36: I'm not sure this conclusion about causation is valid unless the study established that a similar proportion of the MU population was recruited in each condition.

Line 179: I am not at all sure that the prolongation of input by PICs makes them a prime candidate for explaining MUFR firing rate suppression. But perhaps this sentence is meant to imply that PICs are smaller and neuromodulatory drive less? This is what the next para implies.

END OF COMMENTS

Confidential Review

07-Feb-2024

Piasecki JP-TR-2024-284159R1

Response to review

I am grateful to the reviewers and editor for their prompt and constructive review of the manuscript. All comments have been individually addressed below.

EDITOR COMMENTS

Reviewing Editor:

The commissioned review has now been revised by two reviewers who are very expert in the field of neuromuscular control of movement. Both reviewers highlighted pros and cons of the present article. Specifically, Reviewer 1 has provided an interesting and on-point (IMHO) comments related to the conclusions made by the author on how disuse depresses firing rate in low-threshold motor units, whereas it impairs NMJ transmission in high-threshold motor units: "There does not appear to be enough data to conclude that the impact of disuse on NMJs is greater for high-threshold motor units." Further, Rev1 raised some concerns on the quality and clarity of the writing and on the confidence in the conclusions.

The author is invited to carefully address these comments in the next draft of the manuscript.

Many thanks. The reviewer feedback is extremely constructive and in my opinion, has resulted in a much-improved Topical Review.

I agree there is a lack of direct evidence supporting adaptation of MUFR and NMJ instability specifically targeting MUs differing on size/type/recruitment threshold. However, as per my instructions when invited to write the review, and as per JP guidelines here ("*Authors should be forward-looking and present new questions for future research/developments and are encouraged to express their own opinion on a subject area and may be controversial if they wish to be, as science often moves fastest when ideas are challenged*")., I believe there is sufficient evidence to highlight suppression of MUFR with disuse without NMJ instability during lower forces, and numerous histological evidence supporting NMJ disruption for fibres contributing to unknown forces. Bridging the two, whilst also highlighting the lack of evidence combining the two, is my attempt at being '*forward-looking, and presenting new questions for future research*', and comes close to '*expressing my own opinion*'.

As a compromise, I have tempered the language around this issue and made it clear that indirect evidence hints toward this concept, direct evidence is lacking, and future research should and could address this.

I hope the reviewers and editor find this appropriate.

REFEREE COMMENTS

Referee #1:

The purpose of the topical review is to discuss the role of adaptations in the firing rate of motor units and neuromuscular propagation in response to exposure to reduced levels of physical activity. The review focuses on the differences in the adaptations experienced by low- and high-threshold motor units. The conclusion is that disuse depresses firing rate in low-threshold motor units, whereas it impairs NMJ transmission in high-threshold motor units.

General Comments

1. The impact of disuse, such as that associated with leg unloading, varies across muscle and muscle fiber types (see Figure 1 in Fitts et al. JAP 89: 823, 2000).

This is currently a major focus of our wider group, using tibialis anterior and medial gastrocnemius in humans as a model of atrophy Resistant (TA) and atrophy Susceptible (MG) (aRaS) loss with disuse (<https://gtr.ukri.org/projects?ref=BB%2FR010358%2F1>). Our early data indicate this is unlikely to be explained in humans simply by differences in fibre type, as the composition in TA and MG are similar, yet the degree of functional loss and atrophy is not (Bass *et al.*, 2021). Our unpublished data also indicate similar reductions in MUFR in both muscles, as observed in the vastus lateralis from the same study (Inns *et al.*, 2022).

Throughout the manuscript I have specified which muscle group is being referred to.

2. The brief section on NMJs seems to focus mainly on the anatomical adaptations with little information on NMJ, such as that indicated by the stability (e.g., jitter) of neuromuscular propagation. There does not appear to be enough data to conclude that the impact of disuse on NMJs is greater for high-threshold motor units.

The section on NMJ has been expanded (~200 words). As per points above and below, I have tempered the language throughout to highlight direct evidence of disuse preferentially affecting higher threshold MUs, and added several points that support this notion.

Specific Comments

35 Why is it not reasonable to consider the motoneuron pool as a single physiological entity with varying susceptibility to plasticity among its members? For example, low- vs high-

threshold motor units (e.g., Figure 5 in Duchateau & Hainaut, 1990; Figure 5 in Comery et al. J Physiol 568: 841, 2005).

Good point. This is poorly worded and has been amended. It now reads:

“As such, motor units within the pool should be viewed in light of their varying susceptibility to disuse.”

39 What are the black arrows intended to indicate?

The arrows in the abstract figure were intended to indicate transition from lower to higher threshold motor units. This has been redrawn to reflect theoretical and much simplified relationship between motor unit recruitment and muscle force. I appreciate this relationship differs across muscles and contraction types.

58 Contralateral connections within the nervous system suggest that the non-involved limb should not be used as a control (Manca et al. Sports Med 51: 11, 2021).

I agree the opposing limb would serve as a poor control in studies of cross-education which include a training stimulus, as per this Delphi study. However, the available evidence shows minimal detrimental effects of the opposing limb with disuse (Inns et al., 2022; Preobrazenski et al., 2023). To improve clarity I have altered this sentence which now reads:

“Experimental scenarios of individual limb disuse are arguably more tolerant for participants and ethically agreeable, and although limits effects of more systemic disuse, it does enable the exploration of muscle and limb specific adaptation, with the added benefit of using non-immobilised opposing limbs as a comparison (Preobrazenski et al., 2023).”

80 What makes the steps intricate?

That they are composed of multiple complex interrelating parts. But I agree, this is unnecessary wordiness and has been removed.

82 Don't ignore the influence of brainstem pathways on the excitability of motoneurons (Glover & Baker, J Neurosci 42: 3150, 2022).

Good point. This section now reads:

In response to numerous stimuli such as sight, sound, and afferent feedback from proprioceptive impulses, signals from the primary motor cortex (M1) and reticular formation of the brainstem propagate down descending corticospinal and reticulospinal tracts where they are in turn transmitted to interneurons and motoneurons (Kiehn, 2016; Glover & Baker, 2022).”

92 Explain the function served by the two sets of projections.

This section now reads:

“The M1 region in humans and higher primates differs to other animal models and has distinct evolutionary origins. Subdivided into a rostral region (old M1) and a caudal region (new M1), the old may have fewer or slower conducting cortico-motoneuronal cells (Witham et al., 2016) and communicates with lower motoneurons via spinal interneurons, which may not favour detailed movements such as those required for the hand (Rathelot & Strick, 2009).”

95 What aspects of coordination do they control?

96 This paragraph is difficult to read. Perhaps begin the paragraph with a sentence that states the topic to be discussed?

In response to both points, and comments from reviewer 2, it is apparent this section lacked clarity. This has been redrafted as follows:

“Of those descending cells that do not synapse directly onto motoneurons, groups of excitatory and inhibitory interneurons regulate net excitability of motoneurons (Zholudeva et al., 2021). In locomotive movements such as walking, the bilateral regulation of each limb is partly reliant on the prevention of contralateral motoneuron excitation, referred to as crossed inhibition. This can occur directly via inhibitory commissural neurons (CN) or indirectly via excitatory CN acting on premotor inhibitory neurons. Cross excitation can also occur via direct CN excitation of interneurons/motoneurons (Maxwell & Soteropoulos, 2020), and is a prominent consideration for studies of unilateral disuse where effects may occur in the non-immobilised limb.”

104 But some neurotransmitters (e.g., serotonin and noradrenaline) attach to metabotropic receptors.

Good point. In an effort to simplify, this section now reads:

The net excitation of the motoneuron can occur from thousands of synaptic inputs along its dendrites, ranging from descending drive to sensory afferents, all of which must be integrated and regulated. This is achieved via two fundamental mechanisms, ionotropic input and neuromodulation. Post-synaptic ionotropic receptors depolarise or hyperpolarise the cell via regulation of ion exchange across the membrane to generate an excitatory or inhibitory response (Heckman et al., 2009).

107 Neuromodulatory inputs can influence most electrophysiological properties of neurons; that is, more than just recruitment threshold.

Good point. This has been removed.

110 Discuss the influence of these neurotransmitters in terms of gain control (Johnson & Heckman Front Neural Circuits 8: 81, 2014; Johnson et al. J Neurophysiol 118: 520, 2017).

This section has been amended and now reads:

“These neurotransmitters stimulate persistent inward currents (PICs) via G-protein coupled second messengers, which act to amplify and prolong synaptic input to the motoneuron. This enables an input-output gain control of motoneurons that facilitates motor output in response to demand (Johnson & Heckman, 2014).”

113 The ΔF technique provides a measure of the influence of neuromodulatory inputs on sustained firing and not PIC amplitude (Afsharipour et al. J Neurophysiol 124: 63, 2020).

This is a great point. I have adjusted the language to better reflect the ΔF technique. This now reads:

“Direct estimation of PIC amplitude in human spinal motoneurons is not possible, but the well-established ΔF technique, in which MUFR onset-offset hysteresis is calculated during ramped voluntary contractions, is able to estimate the influence of neuromodulatory inputs on sustained MU firing (Gorassini et al., 1998).”

116 Define cluster.

This section has been amended and now reads:

Much of the net excitation is common across multiple MNs and monoaminergic drive is highly diffuse (Johnson & Heckman, 2014), and the selective activation of individual MNs is not possible (Rossato et al., 2024).

123 Del Vecchio

Corrected.

131 Cite the work of Vila-Chã et al. (J Appl Physiol 109: 1455, 2010).

I am familiar with this paper and have cited it many times in previous work, however it does not seem relevant to these points on strength loss with disuse.

133 Replace "suppression" with "depression" throughout the manuscript.

I appreciate this suggestion but prefer to keep suppression. The term denotes an act of restraining or inhibiting, and I believe it fits in this context. We have used this previously in work relevant to this review (Inns et al., 2022) and I am content the term will be recognised and understood. I hope the reviewer agrees.

136 ...cause of the intervention?

Agreed, this is poor wording. This has been amended and now reads:

“Given the pivotal role of MUFR in force generation, this suppression of MUFR post disuse with strength loss may be initially viewed as a consequence of sampling MUs at reduced force levels rather than a cause of lower force generation. Yet the same pattern of suppressed MUFR was apparent when contraction levels were normalised to baseline maximum strength (prior to loss of force), favouring suppressed MUFR as a direct consequence of disuse (Inns et al., 2022).”

150 Alternatively, the depression in MUFR was probably related to the extent the habitual level of activity was reduced by the intervention. For analogous interpretation, see the Figure 6A in Farina et al. (J Neurophysiol 101: 350, 2009) on the adjustments in discharge rate during fatiguing contractions.

This is interesting work on MUFR with fatigue, but I am not sure it is directly relevant to the disuse points raised here. The specific mechanisms of MUFR suppression following disuse remain unknown.

168 But Pearce et al. (Scand J Med Sci Sports 23: 740, 2013) found a decrease in the strength of arm muscles in the contralateral limb after 3 weeks of unilateral immobilization.

My understanding of the Pearce study is that in the immobilised group, only the left arm (the immobilised limb) reduced in strength, with no change in the right (non-immobilised limb). This is shown in Table 2, copied below.

I agree there are additional complexities involving limb dominance in upper vs lower limbs, where interlimb differences may be more apparent in upper limbs.

Table 2. Means (\pm SD) for dynamic strength, isometric force, and muscle thickness of the left (immobilized) and right arm

Group	1-RM strength (kg)		Isometric muscle force (N)		Muscle thickness (mm)	
	Pre	Post	Pre	Post	Pre	Post
Left arm						
Control	15.5 (\pm 1.9)	15.5 (\pm 1.9)	195.8 (\pm 7.5)	196.0 (\pm 8.2)	26.8 (\pm 2.7)	26.9 (\pm 2.7)
Immob	15.1 (\pm 2.5)	12.1 (\pm 1.7)*	198.0 (\pm 8.1)	186.8 (\pm 10.0)*	25.1 (\pm 1.3)	23.6 (\pm 1.5)*
Immob + train	18.0 (\pm 3.2)	17.9 (\pm 3.0)	193.0 (\pm 9.1)	198.2 (\pm 11.6)	23.6 (\pm 2.9)	23.6 (\pm 3.0)
Right arm						
Control	14.1 (\pm 1.9)	14.1 (\pm 1.6)	196.7 (\pm 6.1)	196.4 (\pm 7.7)	25.7 (\pm 1.9)	25.8 (\pm 1.2)
Immob	14.4 (\pm 2.9)	13.8 (\pm 2.2)	198.2 (\pm 6.9)	198.5 (\pm 5.6)	24.9 (\pm 1.4)	24.2 (\pm 1.4)
Immob + train	17.3 (\pm 3.2)	19.7 (\pm 3.4)*	198.5 (\pm 10.5)	210.0 (\pm 9.7)*	24.8 (\pm 2.9)	26.3 (\pm 2.8)*

Asterisk (*) denotes statistical significance (Bonferroni adjusted P -value $<$ 0.016).
1-RM, 1 repetition maximum.

180 But all these studies incorrectly used the ΔF technique to estimate PIC amplitude. To be more precise, these studies show the influence of an intervention on sustained firing.

Great point. This has been redrafted to improve accuracy, and now reads:

“Moreover, the sustained firing of MNs as assessed by ΔF is highly adaptable, with increases in response to increased activity (Orssatto et al., 2023b, 2023a), decreases to induced acute inhibition (Mesquita et al., 2022; Orssatto et al., 2022), and is markedly lower in young males

compared to females (Jenz et al., 2023), and in older compared to young (Hassan et al., 2021; Orssatto et al., 2021; Guo et al., 2024)."

185 Humans only?

Work published in recent months from the same cohort as that referred to throughout this paper (Martino et al., 2024), has now assessed ΔF pre and post limb suspension, albeit with some limitations. I have included coverage of this study throughout.

186 Perhaps phrase this sentence on terms of the control strategy proposed by Johnson et al. (2017)?

This now reads:

"If PICs are altered following disuse, further questions remain as to the causative mechanism; is it reduced descending monoaminergic drive and/or a reduction in monoamine receptors or their impairment, or an increase of inhibitory inputs that influence gain control?"

196 Does this mean that monoaminergic drive is distributed uniformly to each motoneuron pool?

No that was not the intended point, but upon rereading it is evident it is not clear. This section has been redrafted and now reads:

"Considering MUFR in humans were suppressed at multiple force levels (normalised to relative and absolute maximal) after disuse (Inns et al., 2022), it is possible monoaminergic drive did not increase when greater relative effort was required, or MNs were less responsive to it."

Also added in preceding section:

"Much of the net excitation is common across multiple MNs (DeLuca & Erim, 1994) and monoaminergic drive is highly diffuse (Johnson & Heckman, 2014), meaning the selective activation of individual MNs is not possible (Rossato et al., 2024)."

202 Couch this in terms of the decline in habitual levels of activity.

I'm unsure if the reviewer refers to declines in strength and of what constitutes a low or high threshold MU following this adaptation.

205 These parallels are questionable.

Agreed. These points and references have been removed.

207 Not if the issue is presented as a decline in habitual levels of activity.

This sentence has been removed.

209 This comparison seems inappropriate.

I appreciate rodent models can be limited, but this study provides strong evidence of enhanced afferent input in response to disuse, albeit with increased sensitivity to pain only. I think it is a useful comparison.

218 Alternatively, the greater impact on low-threshold motor units may reflect a greater relative loss of excitatory synaptic inputs.

Good point, and I agree evidence is lacking for both. I have removed this final sentence citing probable inhibitory input as the most probable factor.

241 Define nerve:muscle size ratio.

It was intended to signify the size of the motor endplate in relation to the size of its corresponding muscle fibre. Upon rereading, it seems irrelevant to the point and has been removed.

243 Does disruption refer to anatomical properties?

Redrafted to:

"This form of structural NMJ adaptation with disuse is age specific and more evident in those still undergoing development."

244 How is the potency of a disruption quantified?

Changed to 'evident'.

245 What is the translational relevance?

It was intended to illustrate that the concept of young vs old rodents may not translate well to young vs old humans. This phrase has been removed.

247 Explain inhibition of MN activation.

Redrafted to 'prevention of MN activation'.

249 ...was increased size...?

Corrected. Now reads:

'Blocking of the NMJ resulted in decreased size of the synaptic vesicle pool, which was increased when the MN soma was blocked.'

250 How does this work highlight the influence of the level of synaptic input?

Prevention of MN activation via blocking of the soma had distinct effects on NMJ size, which differed from those observed following blocking of the NMJ. Put simply, blocking the 'proximal' region of the MN has opposing effects on the NMJ to blocking the 'peripheral' region. I agree this was not clear and has been amended:

'Nevertheless, the importance of the methods of immobilisation/disuse are underscored using the rodent model, as pharmacological inhibition of the NMJ had opposing effects to the prevention of MN activation via spinal hemi-section (Mantilla et al., 2007); blocking of the NMJ resulted in decreased size of the synaptic vesicle pool, which was increased when the MN soma was blocked (Mantilla et al., 2007). This work shows that MN activation, an unknown quantity in human disuse studies, clearly influences peripheral NMJ adaptation.'

255 Which muscle?

Vastus lateralis. This sentence has been corrected:

"The proportion of NCAM+ fibres in human vastus lateralis increased following three (Demangel et al., 2017), ten (Monti et al., 2021), and fourteen days of bed rest (Arentson-Lantz et al., 2016)"

269 But the described adaptation in NMJ were not evaluated relative to the recruitment thresholds of motor units.

I agree there is no direct evidence of the selective NMJ disruption of higher threshold MUs. Regarding the effects of disuse, electrophysiological evidence supports no NMJ transmission instability on 'lower' threshold MUs, and histological evidence supports structural NMJ adaptation of unknown threshold MUs. I present the possibility of susceptibility to each being relevant to the type of MU (based on recruitment threshold).

This section has been redrafted in an attempt to improve clarity and emphasise it is a hypothesis currently lacking direct evidence. This now reads:

"It is possible that the effects of disuse on MUs are not equally applied to all, with a suppression of MUFR that disproportionately affects lower threshold MUs, and NMJ disruption disproportionately affecting higher threshold MUs".

271 Is there any evidence of such a feedback pathway?

There is not, and referring to it as such would be inaccurate. It is possible increased neural drive is required to compensate for reduced NMJ function. VL MUFR at 50% increased (slightly) following 10 days of limb suspension, and decreased at 10 and 25% MVC (Valli et al., 2023). This section has been amended to improve clarity.

“The two processes are not at odds and are feasibly mutually explanatory; the increase in MUFR of higher threshold MUs employed during higher contractions (50% MVC, (Valli et al., 2023)) may act a compensatory mechanism to overcome NMJ disruption, which does not occur in lower threshold MUs.”

272 susceptibility?

This was poor wording and has been removed.

273 How is the force generated by a single motor unit influenced by that contributed by other motor units?

This was poor wording and has been removed.

275 What is meant by the term "overlap of thresholds of MUs"?

This was poor wording and has been amended. It now reads:

“This is somewhat of a simplification, but the forces in which MUs are sampled should be a key consideration for future mechanistic studies in this field.”

281 These studies do not examine molecular aspects of spinal MNs.

Good point. This was intended to reflect the greater number of MU spike trains that can now be simultaneously sampled. This now reads:

“Molecular aspects of spinal MNs in humans will likely continue to elude us due the huge methodological constraints that prohibit its investigation in vivo. Yet recent advances in both EMG hardware and decomposition techniques are enabling a greater number of MU spike trains, and more importantly, from a greater range of contraction intensities, to be simultaneously sampled across the volume of muscle (Škarabot et al., 2023; Avrillon et al., 2023; Chung et al., 2023).”

291 Limited rather than rigid.

Agreed and amended.

292 Dynamic instead of complex.

Agreed and amended.

293 Inhibition and excitation of what?

Amended to:

“Unilateral MU behaviour during submaximal isometric contractions may reveal little of motor pathway commands during more dynamic functions or those applicable to activities of daily living, such as bilateral MU function during normal gait.”

295 Define computational outputs.

Again, this is poor wording and has been amended:

“This dynamic assessment is methodologically challenging but may generate a research capability in which the MU pool is not viewed as single entity, but as a range with differing susceptibility to intervention.”

Referee #2:

This review addresses a timely and much needed overview of an important topic, changes in neural drive to motoneurons in disuse. It effectively presents a wealth of important results in a thoughtful and thought-provoking manner. I expect it will have a major influence on this field, especially as recent results are really transforming our understanding.

Many thanks for taking the time to review and for the positive overview of this work.

The initial introductory material, however, up to line 127, really needs to be reworked in terms of being more concise and worded more carefully. Once the focus shifts to reviewing recent studies, the wording and phrasing are well done. For the first part, I have noted a few instances of lack of clarity below, but there are certainly others.

Agreed following feedback and rereading. Much of the manuscript, and the majority of this section, has been amended. I believe it is much improved and hope the reviewer agrees.

Much of the initial overview of how motoneurons and MU units function is close to textbook information. This section is crucial, but perhaps review articles would generally be preferable citations instead of single examples of recent studies.

Much of this section has been redrafted, with several references removed and key reviews included.

Line 50: "have occurred" is not needed in this sentence, how about "..disuse have primarily explored.."

Agreed and amended.

Line 51: "This" refers to what exactly in the previous sentence? Perhaps "This focus..."

Amended to:

"These studies have translational implications for clinical settings, spaceflight, and in some instances are used as a model of accelerated ageing."

Line 65: MUF_R is not defined until line 75

Thank you. This is now defined at first use in the preceding section.

Line 76: " its..." should be "their", or MU singular

Agreed. This has been amended and now reads:

"Recruitment of a MU is dependent on its activation threshold which is typically proportional to its size. Smaller motoneurons with higher input resistance reach activation threshold with lower synaptic input compared to larger motoneurons. Smaller MUs are typically composed of fewer and smaller muscle fibres, and generate lower levels of force (Enoka & Duchateau, 2017)."

Line 83: Implying M1 mainly projects to the spinal CPG is odd at best. M1 projects to many types of interneurons and, in primates, directly to motoneurons. The author is clearly aware of this from comments farther down.

Good point. This section now reads:

"In response to numerous stimuli such as sight, sound, and afferent feedback from proprioceptive impulses, signals from the primary motor cortex (M1) and reticular formation of the brainstem propagate down descending corticospinal and reticulospinal tracts where they are in turn transmitted to interneurons and motoneurons (Kiehn, 2016; Glover & Baker, 2022)."

Line 84: "This.." seems to refer to locomotor centers

This section has been shortened in an aim to make the introduction more succinct. It now reads:

"In response to numerous stimuli such as sight, sound, and afferent feedback from proprioceptive impulses, signals from the primary motor cortex (M1) and reticular formation of the brainstem propagate down descending corticospinal and reticulospinal tracts where they are in turn transmitted to the spinal locomotor circuit (Kiehn, 2016; Glover & Baker, 2022). The M1 region in humans and higher primates differs to other animal models and has

distinct evolutionary origins. Subdivided into a rostral region (old M1) and a caudal region (new M1), the old may have fewer or slower conducting cortico-motoneuronal cells (Witham et al., 2016) and communicates with lower motoneurons via spinal interneurons, which may not favour detailed movements such as those required for the hand (Rathelot & Strick, 2009)."

Line 94: "..those interrupted..." is not an appropriate way to describe corticospinal inputs to multiple types of interneurons

Agreed. This now reads:

"Of those descending cells that do not synapse directly onto motoneurons, groups of excitatory and inhibitory interneurons regulate net excitability of motoneurons (Zholudeva et al., 2021)."

Line 109: " in this context" is unnecessary wordiness

Agreed and amended.

Line 117: while Hug et al study is beautiful work establishing "clusters", the study of common drive has a long history and at least a few more references are needed here.

Agreed. This reference has been removed and an earlier DeLuca review is included within the amended text. This now reads:

"Much of the net excitation is common across multiple MNs (DeLuca & Erim, 1994) and monoaminergic drive is highly diffuse (Johnson & Heckman, 2014), meaning the selective activation of individual MNs is not possible (Rossato et al., 2024)."

Line 131: why not stick with the MUFR abbreviation?

Agreed and amended. This now reads:

"MUFR recorded at submaximal contractions (10 and 25% max) was also reduced post-disuse by approximately 10%."

Lines 133-36: I'm not sure this conclusion about causation is valid unless the study established that a similar proportion of the MU population was recruited in each condition.

The proportion of the MU population recruited at each contraction level pre and post disuse was not estimated in this study. Therefore, I have softened the language to indicate the lack of direct causal evidence.

"Given the pivotal role of MUFR in force generation, this suppression of MUFR post disuse may be initially viewed as a consequence of sampling MUs at reduced force levels rather than a cause of lower force generation. Yet the same pattern of suppressed MUFR was

apparent when contraction levels were normalised to baseline maximum strength (prior to loss of force), favouring suppressed MUFR as a direct consequence of disuse (Inns et al., 2022).

Line 179: I am not at all sure that the prolongation of input by PICs makes them a prime candidate for explaining MUFR firing rate suppression. But perhaps this sentence is meant to imply that PICs are smaller and neuromodulatory drive less? This is what the next para implies.

Agreed. Much of this section has been redrafted with a reduced focus on PICs. This also includes findings from a recent study reporting ΔF following disuse (Martino et al., 2024), and of the limitations of that data.

References

- Arentson-Lantz EJ, English KL, Paddon-Jones D & Fry CS (2016). Fourteen days of bed rest induces a decline in satellite cell content and robust atrophy of skeletal muscle fibers in middle-aged adults. *J Appl Physiol* **120**, 965–975.
- Avrillon S, Hug F, Enoka R, Caillet AH & Farina D (2023). *The decoding of extensive samples of motor units in human muscles reveals the rate coding of entire motoneuron pools*. Neuroscience. Available at: <http://biorxiv.org/lookup/doi/10.1101/2023.11.25.568607> [Accessed February 5, 2024].
- Bass JJ, Hardy EJO, Inns TB, Wilkinson DJ, Piasecki M, Morris RH, Spicer A, Sale C, Smith K, Atherton PJ & Phillips BE (2021). Atrophy Resistant vs. Atrophy Susceptible Skeletal Muscles: “aRaS” as a Novel Experimental Paradigm to Study the Mechanisms of Human Disuse Atrophy. *Front Physiol*; DOI: 10.3389/fphys.2021.653060.
- Chung B et al. (2023). *Myomatrix arrays for high-definition muscle recording*. *elife*. Available at: <https://elifesciences.org/reviewed-preprints/88551v2> [Accessed February 5, 2024].
- DeLuca C & Erim Z (1994). Common drive of motor units in regulation of muscle force. *Trends Neurosci* **17**, 299–305.
- Demangel R, Treffel L, Py G, Brioché T, Pagano AF, Bareille M, Beck A, Pessemeesse L, Candau R, Gharib C, Chopard A & Millet C (2017). Early structural and functional signature of 3-day human skeletal muscle disuse using the dry immersion model. *J Physiol* **595**, 4301–4315.
- Enoka RM & Duchateau J (2017). Rate coding and the control of muscle force. *Cold Spring Harb Perspect Med*; DOI: 10.1101/cshperspect.a029702.
- Glover IS & Baker SN (2022). Both Corticospinal and Reticulospinal Tracts Control Force of Contraction. *J Neurosci* **42**, 3150–3164.
- Gorassini MA, Bennett DJ & Yang JF (1998). Self-sustained firing of human motor units. *Neurosci Lett* **247**, 13–16.

Guo Y, Jones EJ, Škarabot J, Inns TB, Phillips BE, Atherton PJ & Piasecki M (2024). Common synaptic inputs and persistent inward currents of vastus lateralis motor units are reduced in older male adults. *GeroScience*; DOI: 10.1007/s11357-024-01063-w.

Hassan AS, Fajardo ME, Cummings M, McPherson LM, Negro F, Dewald JPA, Heckman CJ & Pearcey GEP (2021). Estimates of persistent inward currents are reduced in upper limb motor units of older adults. *J Physiol*; DOI: 10.1113/JP282063.

Heckman CJ, Mottram C, Quinlan K, Theiss R & Schuster J (2009). Motoneuron excitability: The importance of neuromodulatory inputs. *Clin Neurophysiol* **120**, 2040–2054.

Inns TB, Bass JJ, Hardy EJO, Wilkinson DJ, Stashuk DW, Atherton PJ, Phillips BE & Piasecki M (2022). Motor unit dysregulation following 15 days of unilateral lower limb immobilisation. *J Physiol* **600**, 4753–4769.

Jenz ST, Beauchamp JA, Gomes MM, Negro F, Heckman CJ & Pearcey GEP (2023). Estimates of persistent inward currents in lower limb motoneurons are larger in females than in males. *J Neurophysiol* **129**, 1322–1333.

Johnson MD & Heckman CJ (2014). Gain control mechanisms in spinal motoneurons. *Front Neural Circuits*; DOI: 10.3389/fncir.2014.00081.

Kiehn O (2016). Decoding the organization of spinal circuits that control locomotion. *Nat Rev Neurosci* **17**, 224–238.

Mantilla CB, Rowley KL, Zhan W-Z, Fahim MA & Sieck GC (2007). Synaptic vesicle pools at diaphragm neuromuscular junctions vary with motoneuron soma, not axon terminal, inactivity. *Neuroscience* **146**, 178–189.

Martino G, Valli G, Sarto F, Franchi MV, Narici MV & De Vito G (2024). Neuromodulatory Contribution to Muscle Force Production after Short-Term Unloading and Active Recovery. *Med Sci Sports Exerc*; DOI: 10.1249/MSS.0000000000003473.

Maxwell DJ & Soteropoulos DS (2020). The mammalian spinal commissural system: properties and functions. *J Neurophysiol* **123**, 4–21.

Mesquita RNO, Taylor JL, Trajano GS, Škarabot J, Holobar A, Gonçalves BAM & Blazevich AJ (2022). Effects of reciprocal inhibition and whole-body relaxation on persistent inward currents estimated by two different methods. *J Physiol* **600**, 2765–2787.

Monti E, Reggiani C, Franchi MV, Toniolo L, Sandri M, Armani A, Zampieri S, Giacomello E, Sarto F, Sirago G, Murgia M, Nogara L, Marcucci L, Ciciliot S, Šimunic B, Pišot R & Narici MV (2021). Neuromuscular junction instability and altered intracellular calcium handling as early determinants of force loss during unloading in humans. *J Physiol* **599**, 3037–3061.

Orssatto LBR, Blazevich AJ & Trajano GS (2023a). Ageing reduces persistent inward current contribution to motor neurone firing: Potential mechanisms and the role of exercise. *J Physiol* **601**, 3705–3716.

Orssatto LBR, Borg DN, Blazeovich AJ, Sakugawa RL, Shield AJ & Trajano GS (2021). Intrinsic motoneuron excitability is reduced in soleus and tibialis anterior of older adults. *GeroScience*; DOI: 10.1007/s11357-021-00478-z.

Orssatto LBR, Fernandes GL, Blazeovich AJ & Trajano GS (2022). Facilitation–inhibition control of motor neuronal persistent inward currents in young and older adults. *J Physiol* **600**, 5101–5117.

Orssatto LBR, Rodrigues P, Mackay K, Blazeovich AJ, Borg DN, Souza TR de, Sakugawa RL, Shield AJ & Trajano GS (2023b). Intrinsic motor neuron excitability is increased after resistance training in older adults. *J Neurophysiol* **129**, 635–650.

Preobrazenski N, Janssen I & McGlory C (2023). The effects of single-leg disuse on skeletal muscle strength and size in the nonimmobilized leg of uninjured adults: a meta-analysis. *J Appl Physiol* **134**, 1359–1363.

Rossato J, Avrillon S, Tucker K, Farina D & Hug F (2024). The volitional control of individual motor units is constrained within low-dimensional neural manifolds by common inputs. *J Neurosci* 0702242024.

Škarabot J, Beauchamp JA & Pearcey GE (2023). *Human motor unit discharge patterns reveal differences in neuromodulatory and inhibitory drive to motoneurons across contraction levels*. Neuroscience. Available at: <http://biorxiv.org/lookup/doi/10.1101/2023.10.16.562612> [Accessed February 2, 2024].

Valli G, Sarto F, Casolo A, Del Vecchio A, Franchi MV, Narici MV & De Vito G (2023). Lower limb suspension induces threshold-specific alterations of motor units properties that are reversed by active recovery. *J Sport Health Sci* S2095254623000595.

Witham CL, Fisher KM, Edgley SA & Baker SN (2016). Corticospinal Inputs to Primate Motoneurons Innervating the Forelimb from Two Divisions of Primary Motor Cortex and Area 3a. *J Neurosci* **36**, 2605–2616.

Zholudeva LV, Abaira VE, Satkunendrarajah K, McDevitt TC, Goulding MD, Magnuson DSK & Lane MA (2021). Spinal Interneurons as Gatekeepers to Neuroplasticity after Injury or Disease. *J Neurosci* **41**, 845–854.

Dear Mathew,

Re: JP-TR-2024-284159R1 "Neural conditioning to muscle disuse: crossing the threshold from firing rate suppression to neuromuscular junction transmission" by Mathew Piasecki

Thank you for submitting your manuscript to The Journal of Physiology. It has been assessed by a Reviewing Editor and by 2 expert referees and we are pleased to tell you that it is potentially acceptable for publication following satisfactory major revision.

Please address all the points raised and incorporate all requested revisions or explain in your Response to Referees why a change has not been made. We hope you will find the comments helpful and that you will be able to return your revised manuscript within 2 months. If you require longer than this, please contact journal staff: jp@physoc.org. Please note that this letter does not constitute a guarantee for acceptance of your revised manuscript.

REVISION CHECKLIST:

We look forward to receiving your revised submission.

Best wishes,

Laura Bennet
Senior Editor
The Journal of Physiology

EDITOR COMMENTS

Reviewing Editor:

Both reviewers have now read and assessed the new version of this invited review - while one reviewer is satisfied with the new additions and corrections made by the author, one reviewer has highlighted a good amount of potential issues within the article. The author is therefore encouraged to revise the paper once more, and he is also encouraged to integrate the adaptations in the function of neuromuscular junctions with those of motor units in the second part of the manuscript.

REFEREE COMMENTS

Referee #1:

The author has responded positively to most of the comments I provided on the initial version of this commissioned paper. However, I have a few additional points (indicated by line number) for the author to consider.

1 The title is too vague. What is meant by "neural conditioning"? Is the neuromuscular junction really part of the nervous system?

27 Throughout the manuscript, replace "suppressed" with "depressed".

38 The upper limit of motor unit recruitment is approximately 85% of MVC force for most muscles, not the 100% suggested by the dashed line. Also, this upper limit is lower for hand muscles (~60%).

What are the three motoneurons in the abstract figure intended to imply?

68 Motor unit firing rate is not impaired. Rather, the extent of rate modulation is reduced.

91 Classic anatomical studies on descending pathways in primates have established that direct projections are only evident in motoneurons that innervate distal muscles. Instead, the pathways typically terminate on spinal interneurons (Foyssal & Baker, J Physiol 597: 2729, 2019; Hudson et al. J Neurophysiol 113: 937, 2015; Phillips & Porter Prog Brain Res 12: 222, 1964;).

97 The example of walking seems inappropriate. Locomotion is controlled by central pattern generation that intrinsically activate recurrent and reciprocal inhibition pathways that are modulated across different modes of locomotion (e.g., Fig. 33-7 in Kandel et al. Principles of Neural Science, 6th edition) depending on the output from the mesencephalic locomotor region.

143 Use this finding to emphasize your point that rate coding is reduced at low forces but not at moderate forces. However, even at a target force of 25% MVC, the exponential distribution of innervation numbers in a motor unit pool suggests that at least 50% of the pool is recruited to achieve this force. Also, this selective effect on rate coding must involve differential modulation of recruitment and rate coding to produce these different target forces.

153 This interpretation should be couched in terms of the modulation of gain control (Johnson & Heckman, 2014).

159 An important point.

179 The key significance of PICs is the amplification of synaptic input and not the prolongation of MU firing (Johnson et al. J Neurophysiol 118: 520, 2017). As a result, the ΔF measurements provide little information about the impact of PICs. Instead, the effect needs to be assessed based on the initial rate of increase in firing rate (see Fig. 1 in Beauchamp et al. J Neural Eng 20: 016034, 2023).

199 Define "monoaminergic drive".

219 The feedback delivered by group III/IV afferents can be inhibitory or excitatory, depending on the muscle group (Martin et al. J Neurosci 26: 4796, 2006).

273 Does the decline in NMJ transmission stability emerge randomly across muscle fibres or is more correlated among muscle fibers within individual MUs as implied in this sentence?

275 Contractile speed is distributed continuously across a motor unit pool so that some low-threshold motor units have fast twitch contraction times (Fig 2 in Burke & Tsairis Ann NY Acad Sci 228: 145, 1974; Fig 3 in Van Cutsem et al. Can J Appl Physiol 22: 585, 1997).

286 Why would a "compensatory mechanism" be limited to high-threshold motor units? What feedback signal would evoke this response?

Referee #2:

The author has done an impressive job of revising this review. It is now very effective and I have no further comments.

END OF COMMENTS

1st Confidential Review

02-Aug-2024

Piasecki JP-TR-2024-284159R2

Response to review

I am pleased to see the majority of the amendments are satisfactory. Further responses to the additional suggestions are below in red.

All other minor alterations to the manuscript are also highlighted in red.

EDITOR COMMENTS

Reviewing Editor:

Both reviewers have now read and assessed the new version of this invited review - while one reviewer is satisfied with the new additions and corrections made by the author, one reviewer has highlighted a good amount of potential issues within the article. The author is therefore encouraged to revise the paper once more, and he is also encouraged to integrate the adaptations in the function of neuromuscular junctions with those of motor units in the second part of the manuscript.

REFEREE COMMENTS

Referee #1:

The author has responded positively to most of the comments I provided on the initial version of this commissioned paper. However, I have a few additional points (indicated by line number) for the author to consider.

Many thanks. The additional comments are again constructive.

1 The title is too vague. What is meant by "neural conditioning"? Is the neuromuscular junction really part of the nervous system?

This is a fair point. The NMJ would, in my view, be considered part of the nervous system but I can appreciate not all may agree, particularly those focusing on post-synaptic aspects alone. I have changed the title to:

Motor unit adaptation to disuse: crossing the threshold from firing rate suppression to neuromuscular junction transmission.

I prefer to keep the subtitle; I find it dull without.

27 Throughout the manuscript, replace "suppressed" with "depressed".

The same recommendation was made in the previous review round so I can only assume the response was not agreeable. The suggestion is appreciated but I prefer to keep suppression. The term denotes an act of restraining or inhibiting, and I believe it is used appropriately in this context. The term has been used previously and I am content it will be recognised and understood.

38 The upper limit of motor unit recruitment is approximately 85% of MVC force for most muscles, not the 100% suggested by the dashed line. Also, this upper limit is lower for hand muscles (~60%).

What are the three motoneurons in the abstract figure intended to imply?

I also believe that to be true regarding the upper limit of recruitment (although I am not aware of a good reference for it). I believe the central dogma also states this is not true for all muscles and may be as low as 50% MVC in hand muscles. Again, I'm not aware of a good reference for this.

The figure is an “abstract figure” and is intended to briefly summarise the review. The 3 motoneurons in the abstract figure represent MU firing. I have altered the figure to include the probable upper limits of recruitment, and I have included further information in the abstract figure legend.

68 Motor unit firing rate is not impaired. Rather, the extent of rate modulation is reduced.

Good point. This has been changed to “reduction” and now reads:

“Of the data available in humans and animals, a reduction of motor unit firing rate (MUFR) and disruption at the neuromuscular junction (NMJ) are clear consequences of disuse yet underpinning mechanisms are less clear”

91 Classic anatomical studies on descending pathways in primates have established that direct projections are only evident in motoneurons that innervate distal muscles. Instead, the pathways typically terminate on spinal interneurons (Foysal & Baker, J Physiol 597: 2729, 2019; Hudson et al. J Neurophysiol 113: 937, 2015; Phillips & Porter Prog Brain Res 12: 222, 1964;).

This is the point this section aims to make; however it may lack clarity. Now reworded to:

“Subdivided into a rostral region (old M1) and a caudal region (new M1), the old may have fewer or slower conducting cortico-motoneuronal cells and communicates with lower motoneurons via spinal interneurons (Witham *et al.*, 2016).”

97 The example of walking seems inappropriate. Locomotion is controlled by central pattern generation that intrinsically activate recurrent and reciprocal inhibition pathways that are modulated across different modes of locomotion (e.g., Fig. 33-7 in Kandel et al. Principles of

Neural Science, 6th edition) depending on the output from the mesencephalic locomotor region.

This section has been reworded and now reads:

“In bilateral movements, the regulation of each limb is partly reliant on the prevention of contralateral motoneuron excitation, referred to as crossed inhibition.”

143 Use this finding to emphasize your point that rate coding is reduced at low forces but not at moderate forces. However, even at a target force of 25% MVC, the exponential distribution of innervation numbers in a motor unit pool suggests that at least 50% of the pool is recruited to achieve this force. Also, this selective effect on rate coding must involve differential modulation of recruitment and rate coding to produce these different target forces.

Good point. This is indeed used to emphasise the point later in the manuscript.

Line 210 reads:

“A critical point of the available human data is that the reduction in MUFR is threshold specific and 210 preferentially affects lower threshold MUs (Duchateau & Hainaut, 1990; Valli *et al.*, 2023).”

Line 284 reads:

“The two processes are not at odds and are feasibly mutually explanatory; the increase in MUFR of higher threshold MUs employed during higher contractions (50% MVC, (Valli *et al.*, 2023)) may act to overcome impaired muscle contraction caused by NMJ disruption, which does not occur in lower threshold MUs”

I do not follow the calculation to estimate that 50% of the MN pool is active during a 25% MVC contraction and am doubtful this could be effectively proven with current methods. Certainly not in humans. This would of course be dependent on innervation ratio and MU number, and likely differs across muscles. I agree there is a limitation in viewing MUFR independently of recruitment, especially when the same absolute force levels can be achieved pre and post disuse. The total number of MU contributing to force at a given time is difficult to estimate and we are currently restricted to report on only those which we can sample (e.g. larger, superficial).

I have added a short section covering this point. Line 198:

“Notwithstanding this reduced MUFR, these forces post disuse were achievable and presumably, were facilitated via greater MU recruitment. However, current methods limit reporting of MU behaviour to only those which can be sampled.”

153 This interpretation should be couched in terms of the modulation of gain control (Johnson & Heckman, 2014).

A similar comment was made in the previous version, so it is possible I misunderstood the reviewers point when replying. As I understand it, to ‘couch’ is to present with caution and in this case, meaning it is insufficient to mention excitation and inhibition of MNs without also

mentioning gain control, i.e. the process of altering MN excitability to a level appropriate for the task via neuromodulation. This seems sensible and this section has been reworded to:

“At the simplest level, suppression of MUFR in these studies can be viewed as an alteration in the gain control of the MN, and/or an altered balance of excitation and inhibition of MNs, inclusive of the intrinsic properties of MNs; either ionotropic, neuromodulatory, or both (Johnson & Heckman, 2014).”

159 An important point.

Agreed.

179 The key significance of PICs is the amplification of synaptic input and not the prolongation of MU firing (Johnson et al. J Neurophysiol 118: 520, 2017). As a result, the ΔF measurements provide little information about the impact of PICs. Instead, the effect needs to be assessed based on the initial rate of increase in firing rate (see Fig. 1 in Beauchamp et al. J Neural Eng 20: 016034, 2023).

Good point. This has been reworded:

“Recall that monoaminergic (e.g. 5HT and NE) drive to spinal motoneurons has potent effects on the intrinsic excitability of the motoneuron via PIC activation which amplifies synaptic input (Heckman *et al.*, 2009)”

Also altered:

179: “The **onset-offset hysteresis of MN firing** as assessed by ΔF is highly adaptable, with increases in response to increased activity...”

109: “Direct estimation of PIC amplitude in human spinal motoneurons is not possible, but the well-established ΔF technique, in which MUFR onset-offset hysteresis is calculated during ramped voluntary contractions, is able to estimate the influence of neuromodulatory inputs on **amplification of synaptic input** (Gorassini *et al.*, 1998).”

199 Define "monoaminergic drive".

Monoaminergic drive in this context refers to the influence of serotonin and norepinephrine on MN function, but perhaps lacks clarity. The following sections have been amended to remove this phrase:

192: “If the PIC contribution to MN firing are altered following disuse, further questions remain as to the causative mechanism; is it reduced availability of 5HT and NE, and/or a reduction in monoamine receptors or their impairment, or an altered balance of excitatory and inhibitory inputs?”

199: “Furthermore, there is a strong likelihood that 5HT neuronal activity positively correlates with motor output, as shown in cats (Jacobs *et al.*, 2002).”

202: “Considering MUFR in humans were suppressed at multiple force levels (normalised to *relative* and *absolute* maximal) after disuse (Inns *et al.*, 2022), it is possible descending drive did not increase when greater relative effort was required, or MNs were less responsive to it. Notwithstanding this reduced MUFR, these forces post disuse were achievable and presumably, were facilitated via greater MU recruitment. However, current methods limit reporting of MU behaviour to only those which can be sampled.”

219 The feedback delivered by group III/IV afferents can be inhibitory or excitatory, depending on the muscle group (Martin et al. J Neurosci 26: 4796, 2006).

True, but this does not invalidate the point that inflammation may trigger inhibition, as seen in atherogenic muscle inhibition. I have clarified this and reworded this sentence:

“Furthermore, disuse-associated inflammation may also have inhibitory effects via group III/IV inhibitory afferents (Amann, 2012; Jones *et al.*, 2023).”

273 Does the decline in NMJ transmission stability emerge randomly across muscle fibres or is more correlated among muscle fibers within individual MUs as implied in this sentence?

This is unknown and fuels an ongoing debate, with one camp asserting MU loss is a result of MN cell death in the spinal cord, and another asserting denervation is a result of muscle-nerve cross talk occurring in a subset of NMJs, followed by Wallerian degeneration of the axon and eventual MU loss. It is, of course more nuanced and will vary according to underpinning cause (age, disease, trauma). I have reworded this sentence in an aim to be less specific and avoid discussing the unknowns here:

“A possible notable caveat of this process is the susceptibility of some NMJs of higher threshold MUs to undergo disruption while those of lower threshold remain unaltered.”

275 Contractile speed is distributed continuously across a motor unit pool so that some low-threshold motor units have fast twitch contraction times (Fig 2 in Burke & Tsairis Ann NY Acad Sci 228: 145, 1974; Fig 3 in Van Cutsem et al. Can J Appl Physiol 22: 585, 1997).

This is not a consistent finding and the recording of tetanic force of single MUs is a widely used tool employed to identify MU types in animal preparations. In the cited paper, single MUs were identified as fast-fatigable, fast-fatigue-resistant, or slow based on contraction forces and time to peak tension. I appreciate there are complexities in assuming all fibres of a particular type belong to a MU of the same type and all will display similar phenotypic traits, but I believe the comparison to age-related MU remodelling being MU-type specific is valid.

286 Why would a "compensatory mechanism" be limited to high-threshold motor units? What feedback signal would evoke this response?

This may be a poor choice of words. It is intended to highlight a requirement for greater MUFR where muscle contraction is limited due to failure/impairment of NMJ transmission. This has been reworded as follows:

“The two processes are not at odds and are feasibly mutually explanatory; the increase in MUFR of higher threshold MUs employed during higher contractions (50% MVC, (Valli *et al.*, 2023)) may act to overcome impaired muscle contraction caused by NMJ disruption, which does not occur in lower threshold MUs”

Dear Mathew,

Re: JP-TR-2024-284159R2 "Motor unit adaptation to disuse: crossing the threshold from firing rate suppression to neuromuscular junction transmission" by Mathew Piasecki

We are pleased to tell you that your paper has been accepted for publication in The Journal of Physiology.

Authors should note that it is too late at this point to offer corrections prior to proofing. Major corrections at proof stage, such as changes to figures, will be referred to the Editors for approval before they can be incorporated. Only minor changes, such as to style and consistency, should be made at proof stage. Changes that need to be made after proof stage will usually require a formal correction notice.

Best wishes,

Laura Bennet
Senior Editor
The Journal of Physiology

P.S. - You can help your research get the attention it deserves! Check out Wiley's free Promotion Guide for best-practice recommendations for promoting your work at www.wileyauthors.com/eo/guide. You can learn more about Wiley Editing Services which offers professional video, design, and writing services to create shareable video abstracts, infographics, conference posters, lay summaries, and research news stories for your research at www.wileyauthors.com/eo/promotion.

IMPORTANT NOTICE ABOUT OPEN ACCESS: To assist authors whose funding agencies mandate public access to published research findings sooner than 12 months after publication, The Journal of Physiology allows authors to pay an Open Access (OA) fee to have their papers made freely available immediately on publication.

You can check if your funder or institution has a Wiley Open Access Account here: <https://authorservices.wiley.com/author-resources/Journal-Authors/licensing-and-open-access/open-access/author-compliance-tool.html>.

EDITOR COMMENTS

Reviewing Editor:

I have no further comments to add.

Senior Editor:

Nice review, thank you for your submission.

REFeree COMMENTS

Referee #1:

I have no additional comments for the author; however, we do disagree on some points.

2nd Confidential Review

09-Oct-2024